# Learning Variational Temporal Abstraction Embeddings in Option-Induced MDPs

## Abstract

The option framework in hierarchical reinforcement learning has notably advanced the automatic discovery of temporally-extended actions from long-horizon tasks. However, existing methods often struggle with ineffective exploration and unstable updates when learning action and option policies simultaneously. Addressing these challenges, we introduce the Variational Markovian Option Critic (VMOC), an off-policy algorithm with provable convergence that employs variational inference to stabilize updates. VMOC naturally integrates maximum entropy as intrinsic rewards to promote the exploration of diverse and effective options. Furthermore, we adopt low-cost option embeddings instead of traditional, computationally expensive option triples, enhancing scalability and expressiveness. Extensive experiments in challenging Mujoco environments validate VMOC's superior performance over existing on-policy and off-policy methods, demonstrating its effectiveness in learning coherent and diverse option sets suitable for complex tasks.

## 1 Introduction

Recent advancements in deep reinforcement learning (DRL) have demonstrated significant successes across a variety of complex domains, such as mastering the human level of atari [36] and Go [44] games. These achievements underscore the potential of combining reinforcement learning (RL) with powerful function approximators like neural networks [5] to tackle intricate tasks that require nuanced control over extended periods. Despite these breakthroughs, Deep RL still faces substantial challenges, such as insufficient exploration in dynamic environments [18, 13, 42], inefficient learning associated with temporally extended actions [6, 9] and long horizon tasks [30, 4], and vast amounts of samples required for training proficient behaviors [16, 40, 15].

One promising area for addressing these challenges is the utilization of hierarchical reinforcement learning (HRL) [11, 2, 12], a diverse set of strategies that decompose complex tasks into simpler, hierarchical structures for more manageable learning. Among these strategies, the option framework [47], developed on the Semi-Markov Decision Process (SMDP), is particularly effective at segmenting non-stationary task stages into temporally-extended actions known as options. Options are typically learned through a maximum likelihood approach that aims to maximize the expected rewards across trajectories. In this framework, options act as temporally abstracted actions executed over variable time steps, controlled by a master policy that decides when each option should execute and terminate. This structuring not only simplifies the management of complex environments but also enables the systematic discovery and execution of temporal abstractions over long-horizon tasks [24, 23].

However, the underlying SMDP framework is frequently undermined by three key challenges: 1) Insufficient exploration and degradation [20, 37, 23]. As options are unevenly updated using conventional maximum likelihood methods [4, 10, 45, 25, 26], the policy is quickly saturated with early rewarding observations. This typically results in focusing on only low-entropy options that lead

to local optima rewards, causing a single option to either dominate the entire policy or switch every timestep. Such premature convergence limits option diversity significantly. 2) Sample Inefficiency. The semi-Markovian nature inherently leads to sample inefficiency [47, 29]: each policy update at the master level extends over multiple time steps, thus consuming a considerable volume of experience samples with relatively low informational gain. This inefficiency is further exacerbated by the prevalence of on-policy option learning algorithms [4, 52], which require new samples to be collected simultaneously from both high-level master policies and low-level action policies at each gradient step, and thus sample expensive. 3) Computationally expensive. Options are conventionally defined as triples [4] with intra-option policies and termination functions, often modeled using neural networks which are expensive to optimize. These challenges collectively limit the broader adoption and effectiveness of the option framework in real-world scenarios, particularly in complex continuous environments where scalability and stability are critical [14, 34, 26].

To address these challenges, we introduce the Variational Markovian Option Critic (VMOC), a novel off-policy algorithm that integrates the variational inference framework on option-induced MDPs [35]. We first formulate the optimal option-induced SMDP trajectory as a probabilistic inference problem, presenting a theoretical convergence proof of the variational distribution under the soft policy iteration framework [19]. Similar to prior variational methods [31], policy entropy terms naturally arise as intrinsic rewards during the inference procedure. As a result, VMOC not only seeks high-reward options but also maximizes entropy across the space, promoting extensive exploration and maintaining high diversity. We implements this inference procedure as an off-policy soft actor critic [19] algorithm, which allows reusing samples from replay buffer and enhances sample efficiency. Furthermore, to address the computational inefficiencies associated with conventional option triples, we follow [35] and employ low-cost option embeddings rather than complex neural network models. This not only simplifies the training process but also enhances the expressiveness of the model by allowing the agent to capture a more diverse set of environmental dynamics.

Our contributions can be summarized as follows:

- We propose a variational inference approach within the maximum entropy framework to enhance diverse and robust exploration of options.
- We implement an off-policy algorithm that improves sample efficiency.
- We introduce option embeddings into latent variable policies and enhance expressiveness and computational cost-effectiveness of option representations.
- We conduct extensive experiments in OpenAI Gym Mujoco [49] environments, demonstrating that VMOC significantly outperforms other option-based variants in terms of exploration capabilities, sample efficiency, and computational efficiency.

## 2 Preliminary

### 2.1 Control as Structured Variational Inference

Conventionally, the control as inference framework [19, 31, 19, 53] is derived using the maximum entropy objective. In this section, we present an alternative derivation from the perspective of structured variational inference. We demonstrate that this approach provides a more concise and intuitive pathway to the same theoretical results, where the maximum entropy principle naturally emerges through the direct application of variational inference techniques.

Traditional control methods focus on directly maximizing rewards, often resulting in suboptimal trade-offs between exploration and exploitation. By reinterpreting the control problem as a probabilistic inference problem, the control as inference framework incorporates both the reward structure and environmental uncertainty into decision-making, providing a more robust and flexible approach to policy optimization. In this framework, optimality is represented by a binary random variable $\mathcal{E} \in \{0,1\}$[1]. The probability of optimality given a state-action pair $(\mathbf{s}, \mathbf{a})$ is denoted as $P(\mathcal{E} = 1 \mid \mathbf{s}, \mathbf{a}) = \exp(r(\mathbf{s}, \mathbf{a}))$, which is an exponential function of the conventional reward function $r(\mathbf{s}, \mathbf{a})$ that measures the desirability of an action in a specific state. Focusing on $\mathcal{E} = 1$ captures the occurrence of optimal events. For simplicity, we will use $\mathcal{E}$ instead of $\mathcal{E} = 1$ in the following text

---

[1]Conventionally, the optimality variable is denoted by $\mathcal{O}$. However, in this context, we use $\mathcal{E}$ to avoid conflict with notation used in the option framework.

to avoid cluttered notations. The joint distribution over trajectories $\tau = (\mathbf{s}_1, \mathbf{a}_1, \ldots, \mathbf{s}_T, \mathbf{a}_T)$ given optimality is expressed as:

$$P(\tau|\mathcal{E}_{1:T}) \propto P(\tau, \mathcal{E}_{1:T}) = P(\mathbf{s}_1) \prod_{t=1}^{T-1} P(\mathbf{s}_{t+1}|\mathbf{s}_t, \mathbf{a}_t) P(\mathcal{E}_t|\mathbf{s}_t, \mathbf{a}_t)$$

where $P(\mathbf{s}_1)$ is the initial state distribution, $P(\mathbf{s}_{t+1}|\mathbf{s}_t, \mathbf{a}_t)$ is the dynamics model. As explained in [19, 31], direct optimization of $P(\tau \mid \mathcal{E}_{1:T})$ can result in an optimistic policy that assumes a degree of control over the dynamics. One way to correct this risk-seeking behavior [31] is through structured variational inference. In our case, the goal is to approximate the optimal trajectory $P(\tau)$ with the variational distribution:

$$q(\tau) = P(\mathbf{s}_1) \prod_{t=1}^{T-1} P(\mathbf{s}_{t+1} \mid \mathbf{s}_t, \mathbf{a}_t) q(\mathbf{a}_t \mid \mathbf{s}_t)$$

where the initial distribution $P(\mathbf{s}_1)$ and transition distribution $P(\mathbf{s}_{t+1} \mid \mathbf{s}_t, \mathbf{a}_t)$ is set to be the true environment dynamics from $P(\tau)$. The only variational term is the variational policy $q(\mathbf{a}_t \mid \mathbf{s}_t)$, which is used to approximate the optimal policy $P(\mathbf{a}_t \mid \mathbf{s}_t, \mathcal{E}_{1:T})$. Under this setting, the environment dynamics will be canceled out from the optimization objective between $P(\tau \mid \mathcal{E})$ and $q(\tau)$, thus explicitly disallowing the agent to influence its dynamics and correcting the risk-seeking behavior.

With the variational distribution at hand, the conventional maximum entropy framework can be recovered through a direct application of standard structural variational inference [28]:

$$\log P(\mathcal{E}_{1:T}) = \mathcal{L}(q(\tau), P(\tau, \mathcal{E}_{1:T})) + D_{\mathrm{KL}}(q(\tau) \parallel P(\tau|\mathcal{E}_{1:T}))$$

$$= \underbrace{\mathbb{E}_{\tau \sim q(\tau)}[\sum_t r(\mathbf{s}_t, \mathbf{a}_t) + \mathcal{H}(q(\cdot|\mathbf{s}_t))]}_{\text{maximum entropy objective}} + D_{\mathrm{KL}}(q(\mathbf{a}_t|\mathbf{s}_t) \parallel P(\mathbf{a}_t|\mathbf{s}_t, \mathcal{E}_{1:T}))$$

where $\mathcal{L}(q, P) = \mathbb{E}_q[\log \frac{P}{q}]$ is the Evidence Lower Bound (ELBO) [28]. The maximum entropy objective arises naturally as the environment dynamics in $P(\tau, \mathcal{E})$ and $q(\tau)$ cancel out. Under this formulation, the soft policy iteration theorem [19] has an elegant Expectation-Maximization (EM) algorithm [28] interpretation: the E-step corresponds to the policy evaluation of the maximum entropy objective $\mathcal{L}(q^{[k]}, P)$; while the M-step corresponds to the policy improvement of the $D_{\mathrm{KL}}$ term $q^{[k+1]} = \arg\max_q D_{\mathrm{KL}}(q^{[k]}(\tau) \parallel P(\tau \mid \mathcal{E}))$. Thus, soft policy iteration is an exact inference if both EM steps can be performed exactly.

**Theorem 1** (Convergence Theorem for Soft Policy Iteration). *Let $\tau$ be the latent variable and $\mathcal{E}$ be the observed variable. Define the variational distribution $q(\tau)$ and the log-likelihood $\log P(\mathcal{E})$. Let $M : q^{[k]} \to q^{[k+1]}$ represent the mapping defined by the EM steps inference update, so that $q^{[k+1]} = M(q^{[k]})$. The likelihood function increases at each iteration of the variational inference algorithm until convergence conditions are satisfied.*

*Proof.* See Appendix A.1. □

## 2.2 The Option Framework

In conventional SMDP-based Option Framework [47], an option is a triple $(\mathbb{I}_o, \pi_o, \beta_o) \in \mathcal{O}$, where $\mathcal{O}$ denotes the option set; $o \in \mathbb{O} = \{1, 2, \ldots, K\}$ is a positive integer index which denotes the $o$-th triple where $K$ is the number of options; $\mathbb{I}_o$ is an initiation set indicating where the option can be initiated; $\pi_o = P_o(\mathbf{a}|\mathbf{s}) : \mathbb{A} \times \mathbb{S} \to [0, 1]$ is the action policy of the $o$th option; $\beta_o = P_o(\mathbf{b} = 1|\mathbf{s}) : \mathbb{S} \to [0, 1]$ where $\mathbf{b} \in \{0, 1\}$ is a *termination function*. For clarity, we use $P_o(\mathbf{b} = 1|\mathbf{s})$ instead of $\beta_o$ which is widely used in previous option literatures (e.g., Sutton et al. [47], Bacon et al. [4]). A *master policy* $\pi(\mathbf{o}|\mathbf{s}) = P(\mathbf{o}|\mathbf{s})$ where $\mathbf{o} \in \mathbb{O}$ is used to sample which option will be executed. Therefore, the dynamics (stochastic process) of the option framework is written as:

$$P(\tau) = P(\mathbf{s}_0, \mathbf{o}_0) \prod_{t=1}^{\infty} P(\mathbf{s}_t|\mathbf{s}_{t-1}, \mathbf{a}_{t-1}) P_{o_t}(\mathbf{a}_t|\mathbf{s}_t)$$

$$[P_{o_{t-1}}(\mathbf{b}_t = 0|\mathbf{s}_t)\mathbf{1}_{\mathbf{o}_t = o_{t-1}} + P_{o_{t-1}}(\mathbf{b}_t = 1|\mathbf{s}_t)P(\mathbf{o}_t|\mathbf{s}_t)], \qquad (1)$$

where $\tau = \{\mathbf{s}_0, \mathbf{o}_0, \mathbf{a}_0, \mathbf{s}_1, \mathbf{o}_1, \mathbf{a}_1, \ldots\}$ denotes the trajectory of the option framework. $\mathbf{1}$ is an indicator function and is only true when $\mathbf{o}_t = o_{t-1}$ (notice that $o_{t-1}$ is the realization at $\mathbf{o}_{t-1}$). Therefore, under this formulation the option framework is defined as a Semi-Markov process since the dependency on an activated option $o$ can cross a variable amount of time [47]. Due to the nature of SMDP assumption, conventional option framework is unstable and computationally expensive to optimize. Li et al. [34, 35] proposed the Hidden Temporal Markovian Decision Process (HiT-MDP):

$$P(\tau) = P(\mathbf{s}_0, \mathbf{o}_0) \prod_{t=1}^{\infty} P(\mathbf{s}_t|\mathbf{s}_{t-1}, \mathbf{a}_{t-1}) P(\mathbf{a}_t|\mathbf{s}_t, \mathbf{o}_t) P(\mathbf{o}_t|\mathbf{s}_t, \mathbf{o}_{t-1}) \tag{2}$$

and theoretically proved that the option-induced HiT-MDP is homomorphically equivalent to the conventional SMDP-based option framework. Following RL conventions, we use $\pi^A = P(\mathbf{a}_t|\mathbf{s}_t, \mathbf{o}_t)$ to denote the action policy and $\pi^O = P(\mathbf{o}_t|\mathbf{s}_t, \mathbf{o}_{t-1})$ to denote the option policy respectively. In HiT-MDPs, options can be viewed as latent variables with a temporal structure $P(\mathbf{o}_t|\mathbf{s}_t, \mathbf{o}_{t-1})$, enabling options to be represented as dense latent embeddings rather than traditional option triples. They demonstrated that learning options as embeddings on HiT-MDPs offers significant advantages in performance, scalability, and stability by reducing variance. However, their work only derived an on-policy policy gradient algorithm for learning options on HiT-MDPs. In this work, we extend their approach to an off-policy algorithm under the variational inference framework, enhancing exploration and sample efficiency.

## 3 Methodology

In this section, we introduce the Variational Markovian Option Critic (VMOC) algorithm by extending the variational policy iteration (Theorem 1) to the option framework. In Section 3.1, we reformulate the optimal option trajectory and the variational distribution as probabilistic graphical models (PGMs), propose the corresponding variational objective, and present a provable exact inference procedure for these objectives in tabular settings. Section 3.2 extends this result by introducing VMOC, a practical off-policy option learning algorithm that uses neural networks as function approximators and proves the convergence of VMOC under approximate inference settings. Our approach differs from previous works [19, 33, 34] by leveraging structured variational inference directly, providing a more concise pathway to both theoretical results and practical algorithms.

### 3.1 PGM Formulations of The Option Framework

Formulating complex problems as probabilistic graphical models (PGMs) offers a consistent and flexible framework for deriving principled objectives, analyzing convergence, and devising practical algorithms. In this section, we first formulate the optimal trajectory of the conventional SMDP-based option framework (Eq. 1) as a PGM. We then use the HiT-MDPs as the variational distribution to approximate this optimal trajectory. With these PGMs, we can straightforwardly derive the variational objective, where maximum entropy terms arise naturally. This approach allows us to develop a stable algorithm for learning diversified options and preventing degeneracy. Specifically, we follow [31, 28]

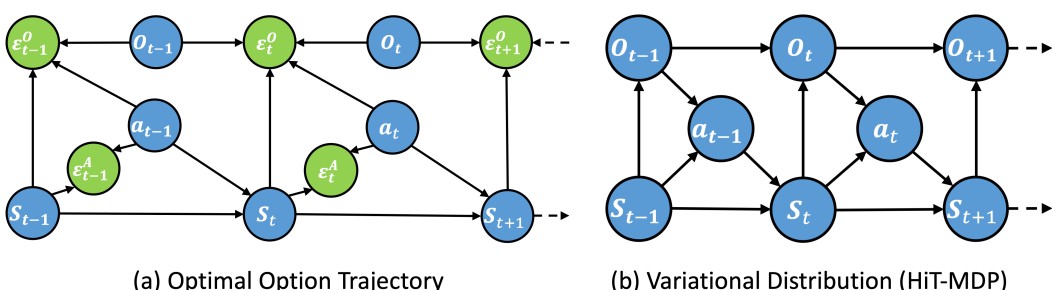

(a) Optimal Option Trajectory          (b) Variational Distribution (HiT-MDP)

**Figure 1:** PGMs of the option framework.

by introducing the concept of "Optimality" [48] into the conventional SMDP-based option framework (Equation equation 1). This allows us to define the probability of an option trajectory being optimal

as a probabilistic graphical model (PGM), as illustrated in Figure 1 (a):

$$P(\tau, \mathcal{E}_{1:T}^A, \mathcal{E}_{1:T}^O) = P(\mathbf{s}_0, \mathbf{o}_0) \prod_{t=1}^{T} P(\mathbf{s}_{t+1}|\mathbf{s}_t, \mathbf{a}_t) P(\mathcal{E}_t^A = 1|\mathbf{s}_t, \mathbf{a}_t) P(\mathcal{E}_t^O = 1|\mathbf{s}_t, \mathbf{a}_t, \mathbf{o}_t, \mathbf{o}_{t-1}) P(\mathbf{o}_t) P(\mathbf{a}_t)$$

$$\propto P(\mathbf{s}_0) \underbrace{\prod_{t=1}^{T} P(\mathbf{s}_{t+1}|\mathbf{s}_t, \mathbf{a}_t)}_{\text{Environment Dynamics}} \underbrace{\prod_{t=1}^{T} P(\mathcal{E}_t^A = 1|\mathbf{s}_t, \mathbf{a}_t) P(\mathcal{E}_t^O = 1|\mathbf{s}_t, \mathbf{a}_t, \mathbf{o}_t, \mathbf{o}_{t-1})}_{\text{Optimality Likelihood}}, \quad (3)$$

where $\mathcal{E} \in \{0, 1\}$ are observable binary "optimal random variables" [31], $\tau = \{\mathbf{s}_0, \mathbf{o}_0, \mathbf{a}_0, \mathbf{s}_1 \ldots\}$ denotes the trajectory of the option framework. The agent is *optimal* at time step $t$ when $P(\mathcal{E}_t^A = 1|\mathbf{s}_t, \mathbf{a}_t)$ and $P(\mathcal{E}_t^O = 1|\mathbf{s}_t, \mathbf{a}_t, \mathbf{o}_t, \mathbf{o}_{t-1})$. We will use $\mathcal{E}$ instead of $\mathcal{E} = 1$ in the following text to avoid cluttered notations. To simplify the derivation, priors $P(\mathbf{o})$ and $P(\mathbf{a})$ can be assumed to be uniform distributions without loss of generality [31]. Note that Eq. 3 shares the same environment dynamics with Eq. 1 and Eq. 2. With the optimal random variables $\mathcal{E}^O$ and $\mathcal{E}^A$, the likelihood of a state-action $\{\mathbf{s}_t, \mathbf{a}_t\}$ pair that is optimal is defined as:

$$P(\mathcal{E}_t^A|\mathbf{s}_t, \mathbf{a}_t) = \exp(r(\mathbf{s}_t, \mathbf{a}_t)), \quad (4)$$

as this specific design facilitates recovering the value function at the latter structural variational infer- ence stage. Based on the same motivation, the likelihood of an option-state-action $\{\mathbf{o}_t, \mathbf{s}_t, \mathbf{a}_t, \mathbf{o}_{t-1}\}$ pair that is optimal is defined as,

$$P(\mathcal{E}_t^O|\mathbf{s}_t, \mathbf{a}_t, \mathbf{o}_{t-1}) = \exp(f(\mathbf{o}_t, \mathbf{s}_t, \mathbf{a}_t, \mathbf{o}_{t-1})), \quad (5)$$

where $f(\cdot)$ is an arbitrary non-positive function which measures the preferable of selecting an option given state-action pair $[\mathbf{s}_t, \mathbf{a}_t]$ and the previous executed option $\mathbf{o}_{t-1}$. In this work, we choose $f$ to be the mutual-information $f = I[\mathbf{o}_t|\mathbf{s}_t, \mathbf{a}_t, \mathbf{o}_{t-1}]$ as a fact that when the uniform prior assumption of $P(\mathbf{o})$ is relaxed the optimization introduces a mutual-information as a regularizer [35].

As explained in Section 2.1, direct optimization of Eq. 3 results in optimistic policies that assumes a degree of control over the dynamics. We correct this risk-seeking behavior [31] through approximating the optimal trajectory $P(\tau)$ with the variational distribution:

$$q(\tau) = P(\mathbf{s}_0, \mathbf{o}_0) \prod_{t=1}^{T-1} P(\mathbf{s}_{t+1}|\mathbf{s}_t, \mathbf{a}_t) q(\mathbf{a}_t|\mathbf{s}_t, \mathbf{o}_t) q(\mathbf{o}_t|\mathbf{s}_t, \mathbf{o}_{t-1}) \quad (6)$$

where the initial distribution $P(\mathbf{s}_0, \mathbf{o}_0)$ and transition distribution $P(\mathbf{s}_{t+1} \mid \mathbf{s}_t, \mathbf{a}_t)$ is set to be the true environment dynamics from $P(\tau)$. The variational distribution turns out to be the HiT-MDP, where the action policy $q(\mathbf{a}_t \mid \mathbf{s}_t)$ and the option policy $q(\mathbf{o}_t|\mathbf{s}_t, \mathbf{o}_{t-1})$ are used to approximate the optimal policy $P(\mathbf{a}_t|\mathbf{s}_t, \mathbf{o}_t, \mathcal{E}_{1:T}^A)$ and $P(\mathbf{o}_t|\mathbf{s}_t, \mathbf{o}_{t-1}, \mathcal{E}_{1:T}^O)$. The Evidence Lower Bound (ELBO) [28] of the log-likelihood optimal trajectory (Eq. 3) can be derived as (see Appendix A.3):

$$\begin{aligned} \mathcal{L}(q(\tau), P(\tau, \mathcal{E}_{1:T}^A, \mathcal{E}_{1:T}^O)) &= \mathbb{E}_{q(\tau)}[\log P(\tau, \mathcal{E}_{1:T}^A, \mathcal{E}_{1:T}^O) - \log q(\tau)] \\ &= \mathbb{E}_{q(\tau)}[r(\mathbf{s}_t, \mathbf{a}_t) + f(\cdot) - \log q(\mathbf{a}_t|\mathbf{s}_t, \mathbf{o}_t) - \log q(\mathbf{o}_t|\mathbf{s}_t, \mathbf{o}_{t-1})] \\ &= \mathbb{E}_{q(\tau)}\left[r(\mathbf{s}_t, \mathbf{a}_t) + f(\cdot) + \mathcal{H}[\pi^A] + \mathcal{H}[\pi^O]\right] \end{aligned} \quad (7)$$

where line 2 is substituting Eq. 3 and Eq. 6 into the ELBO. As a result, the maximum entropy objective naturally arises in Eq. 7. Optimizing the ELBO not only seeks high-reward options but also maximizes entropy across the space, promoting extensive exploration and maintaining high diversity.

Given the ELBO, we now define soft value functions of the option framework following the Bellman Backup Functions along the trajectory $q(\tau)$ as bellow:

$$Q_O^{soft}[\mathbf{s}_t, \mathbf{o}_t] = f(\cdot) + \mathbb{E}_{\mathbf{a}_t \sim \pi^A}\left[Q_A^{soft}[\mathbf{s}_t, \mathbf{o}_t, \mathbf{a}_t]\right] + H[\pi^A], \quad (8)$$

$$Q_A^{soft}[\mathbf{s}_t, \mathbf{o}_t, \mathbf{a}_t] = r(s, a) + \mathbb{E}_{\mathbf{s}_{t+1} \sim P(\mathbf{s}_{t+1}|\mathbf{s}_t, \mathbf{a}_t)}\left[\mathbb{E}_{\mathbf{o}_{t+1} \sim \pi^O}\left[Q_O^{soft}[\mathbf{s}_{t+1}, \mathbf{o}_{t+1}]\right] + H[\pi^O]\right] \quad (9)$$

Assuming policies $\pi^A, \pi^O \in \Pi$ where $\Pi$ is an arbitrary feasible set, under a tabular setting where the inference on $\mathcal{L}$ can be done exactly, we have the following theorem holds:

**Theorem 2** (Soft Option Policy Iteration Theorem). *Repeated optimizing $\mathcal{L}$ and $D_{\mathrm{KL}}$ defined in Eq. 10 from any $\pi_0^A, \pi_0^O \in \Pi$ converges to optimal policies $\pi^{A*}, \pi^{O*}$ such that $Q_O^{soft*}[\mathbf{s}_t, \mathbf{o}_t] \geq Q_O^{soft}[\mathbf{s}_t, \mathbf{o}_t]$ and $Q_A^{soft*}[\mathbf{s}_t, \mathbf{o}_t, \mathbf{a}_t] \geq Q_A^{soft}[\mathbf{s}_t, \mathbf{o}_t, \mathbf{a}_t]$, for all $\pi_0^A, \pi_0^O \in \Pi$ and $(\mathbf{s}_t, \mathbf{a}_t, \mathbf{o}_t) \in \mathcal{S} \times \mathcal{A} \times \mathcal{O}$, assuming under tabular settings where $|\mathcal{S}| < \infty, |\mathcal{O}| < \infty, |\mathcal{A}| < \infty$.*

*Proof.* See Appendix A.2. □

Theorem 2 guarantees finding the optimal solution only when the inference can be done exactly under tabular settings. However, real-world applications often involve large continuous domains and employ neural networks as function approximators. In these cases, inference procedures can only be done approximately. This necessitate a practical approximation algorithm which we present below.

## 3.2 Variational Markovian Option Critic Algorithm

Formulating complex problems as probabilistic graphical models (PGMs) allowing us to leverage established methods from PGM literature to address the associated inference and learning challenges in real-world applications. To this end, we utilizes the structured variational inference treatment for optimizing the log-likelihood of optimal trajectory and prove its convergence under approximate inference settings. Specifically, using the variational distribution $q(\tau)$ (Eq. 6) as an approximator, the ELBO can be derived as (see Appendix A.3):

$$\mathcal{L}(q(\tau), P(\tau, \mathcal{E}_{1:T}^A, \mathcal{E}_{1:T}^O)) = -D_{\mathrm{KL}}(q(\tau)||P(\tau|\mathcal{E}_{1:T}^A, \mathcal{E}_{1:T}^O)) + \log P(\mathcal{E}_{1:T}^A, \mathcal{E}_{1:T}^O) \quad (10)$$

where $D_{\mathrm{KL}}$ is the KL-Divergence between the trajectory following variational policies $q(\tau)$ and optimal policies $P(\tau|\mathcal{E}_{1:T}^A, \mathcal{E}_{1:T}^O)$. Under the structural variational inference [28] perspective, convergence to the optimal policy can be achieved by optimizing the ELBO with respect to the the variational policy repeatedly:

**Theorem 3** (Convergence Theorem for Variational Markovian Option Policy Iteration). *Let $\tau$ be the latent variable and $\mathcal{E}^A, \mathcal{E}^O$ be the ground-truth optimality variables. Define the variational distribution $q(\tau)$ and the true log-likelihood of optimality $\log P(\mathcal{E}^A, \mathcal{E}^O)$. iterates according to the update rule $q^{k+1} = \arg\max_q \mathcal{L}(q(\tau), P(\tau, \mathcal{E}_{1:T}^A, \mathcal{E}_{1:T}^O))$ converges to the maximum value bounded by the true log-likelihood of optimality.*

*Proof.* See Appendix A.4. □

We further implements a practical algorithm, the Variational Markovian Option Critic (VMOC) algorithm, which is suitable for complex continuous domains. Specifically, we employ parameterized neural networks as function approximators for both the Q-functions ($Q_{\psi^A}^{soft}, Q_{\psi^O}^{soft}$) and the policies ($\pi_{\theta^A}, \pi_{\theta^O}$). Instead of running evaluation and improvement to full convergence using Theorem 2, we can optimize the variational distribution by taking stochastic gradient descent following Theorem 3 with respect to the ELBO (Eq. 7) directly. Share the same motivation with Haarnoja et al. [19] of reducing the variance during the optimization procedure, we derive an option critic framework by optimizing the maximum entropy objectives between the action Eq. 9 and the option Eq. 8 alternatively. The Bellman residual for the action critic is:

$$J_{Q^A}(\psi_i^A) = \mathbb{E}_{(\mathbf{s}_t, \mathbf{o}_t, \mathbf{a}_t, \mathbf{s}_{t+1}) \sim D} \left[ \left( \min_{i=1,2} Q_{\psi_i^A}(\mathbf{s}_t, \mathbf{o}_t, \mathbf{a}_t) - \right.\right.$$
$$\left.\left. \left( r(\mathbf{s}_t, \mathbf{a}_t) + \mathbb{E}_{\mathbf{o}_{t+1} \sim \pi^O} \left[ Q_O^{soft}[\mathbf{s}_{t+1}, \mathbf{o}_{t+1}] \right] + \alpha^O H[\pi^O] \right) \right)^2 \right]$$

where $\alpha^O$ is the temperature hyper-parameter and the expectation over option random variable $\mathbb{E}_{\mathbf{o}_{t+1} \sim \pi^O}$ can be evaluated exactly since $\pi^O$ is a discrete distribution. The Bellman residual for the option critic is:

$$J_{Q^O}(\psi_i^O) = \mathbb{E}_{(\mathbf{s}_t, \mathbf{o}_t) \sim D} \left[ \left( \min_{i=1,2} Q_{\psi_i^O}^O(\mathbf{s}_t, \mathbf{o}_t) - \right.\right.$$
$$\left.\left. \left( f(\cdot) + \mathbb{E}_{\mathbf{a}_t \sim \pi^A} \left[ Q_A^{soft}[\mathbf{s}_t, \mathbf{o}_t, \mathbf{a}_t] - \alpha^A \log q(\mathbf{a}_t|\mathbf{s}_t, \mathbf{o}_t) \right] \right) \right)^2 \right]$$

$\alpha^A$ is the temperature hyper-parameter. Unlike $\mathbb{E}_{\mathbf{o}_{t+1} \sim \pi^O}$ can be trivially evaluated, evaluating $\mathbb{E}_{\mathbf{a}_t \sim \pi^A}$ is typically intractable. Therefore, in implementation we use $\mathbf{a}_t$ sampled from the replay buffer to estimate the expectation over $\pi^A$.

Following Theorem 3, the policy gradients can be derived by directly taking gradient with respect to the ELBOs defined for the action Eq. 9 and the option Eq. 8 policies respectively. The action policy objective is given by:

$$J_{\pi^A}(\theta^A) = -\mathbb{E}_{(\mathbf{s}_t, \mathbf{o}_t) \sim D} \left[ \min_{i=1,2} Q_{\psi_i^A}(\mathbf{s}_t, \mathbf{o}_t, \tilde{\mathbf{a}}_t) - \alpha^A \log q(\tilde{\mathbf{a}}_t | \mathbf{s}_t, \mathbf{o}_t) \right], \; \tilde{\mathbf{a}}_t \sim q(\cdot | \mathbf{s}_t, \mathbf{o}_t)$$

where in practice the action policy is often sampled by using the re-parameterization trick introduced in [19]. The option objective is given by:

$$J_{\pi^O}(\theta^O) = -\mathbb{E}_{(\mathbf{s}_t, \mathbf{o}_{t-1}) \sim D} \left[ \min_{i=1,2} Q_{\psi_i^O}(\mathbf{s}_t, \mathbf{o}_t) + \alpha^O \mathcal{H}[\pi^O] \right]$$

The variational distribution $q(\tau)$ defined in Eq. 6 allows us to learn options as embeddings [34, 35] with a learnable embedding matrix $\mathbf{W} \in \mathbb{R}^{\text{num\_options} \times \text{embedding\_dim}}$. Under this setting, the embedding matrix $\mathbf{W}$ can be absorbed into the parameter vector $\theta^O$. This integration into VMOC ensures that options are represented as embeddings without any additional complications, thereby enhancing the expressiveness and scalability of the model.

The temperature hyper-parameters can also be adjusted by minimizing the following objective:

$$J(\alpha^A) = -\mathbb{E}_{\tilde{\mathbf{a}}_t \sim \pi^A} \left[ \alpha^A (\log \pi^A(\tilde{\mathbf{a}}_t \mid \mathbf{s}_t, \mathbf{o}_t) + \overline{\mathcal{H}}) \right]$$

for the action policy temperature $\alpha^A$, where $\overline{\mathcal{H}}$ is a target entropy. Similarly, the option policy temperature $\alpha^O$ can be adjusted by:

$$J(\alpha^O) = -\mathbb{E}_{\mathbf{o}_t \sim \pi^O} \left[ \alpha^O (\log \pi^O(\mathbf{o}_t \mid \mathbf{s}_t, \mathbf{o}_{t-1}) + \overline{\mathcal{H}}) \right]$$

where $\overline{\mathcal{H}}$ is also a target entropy for the option policy. In both cases, the temperatures $\alpha^A$ and $\alpha^O$ are updated using gradient descent, ensuring that the entropy regularization terms dynamically adapt to maintain a desired level of exploration. This approach aligns with the methodology proposed in SAC [19]. By adjusting the temperature parameters, the VMOC algorithm ensures a balanced trade-off between exploration and exploitation, which is crucial for achieving optimal performance in complex continuous control tasks. We summarize the VMOC algorithm in Appendix B.

## 4   Experiments

In this section, we design experiments on the challenging single task OpenAI Gym MuJoCo [7] environments (10 environments) to test Variational Markovian Option Critic (VMOC)'s performance over other option variants and non-option baselines.

For VMOC in all environments, we fix the temperature rate for both $\alpha^O$ and $\alpha^A$ to 0.05; we add an exploration noise $\mathcal{N}(\mu = 0, \sigma = 0.2)$ during exploration. For all baselines, we follow DAC [52]'s open source implementations and compare our algorithm with six baselines, five of which are option variants, *i.e.*, MOPG [35], DAC+PPO, AHP+PPO [32], IOPG [45], PPOC [27], OC [4] and PPO [41]. All baselines' parameters used by DAC remain unchanged over 1 million environment steps to converge. Figures are plotted following DAC's style: curves are averaged over 10 independent runs and smoothed by a sliding window of size 20. Shaded regions indicate standard deviations. All experiments are run on an Intel® Core™ i9-9900X CPU @ 3.50GHz with a single thread and process. Our implementation details are summarized in Appendix C. For a fair comparison, we follow option literature conventions and use four options in all implementations. Our code is available in supplemental materials.

## 5   Experiments

We evaluate the performance of VMOC against six option-based baselines (MOPG [35], DAC+PPO [52], AHP+PPO [32], IOPG [45], PPOC [27], and OC [4]) as well as the hierarchy-free

PPO algorithm [41]. Previous studies [27, 45, 20, 52] have suggested that option-based algorithms do not exhibit significant advantages over hierarchy-free algorithms in single-task environments. Nonetheless, our results demonstrate that VMOC significantly outperforms all baselines in terms of episodic return, convergence speed, step variance, and variance across 10 runs, as illustrated in Figure 2. The only exception is the relatively simple InvertedDoublePendulum environment, which

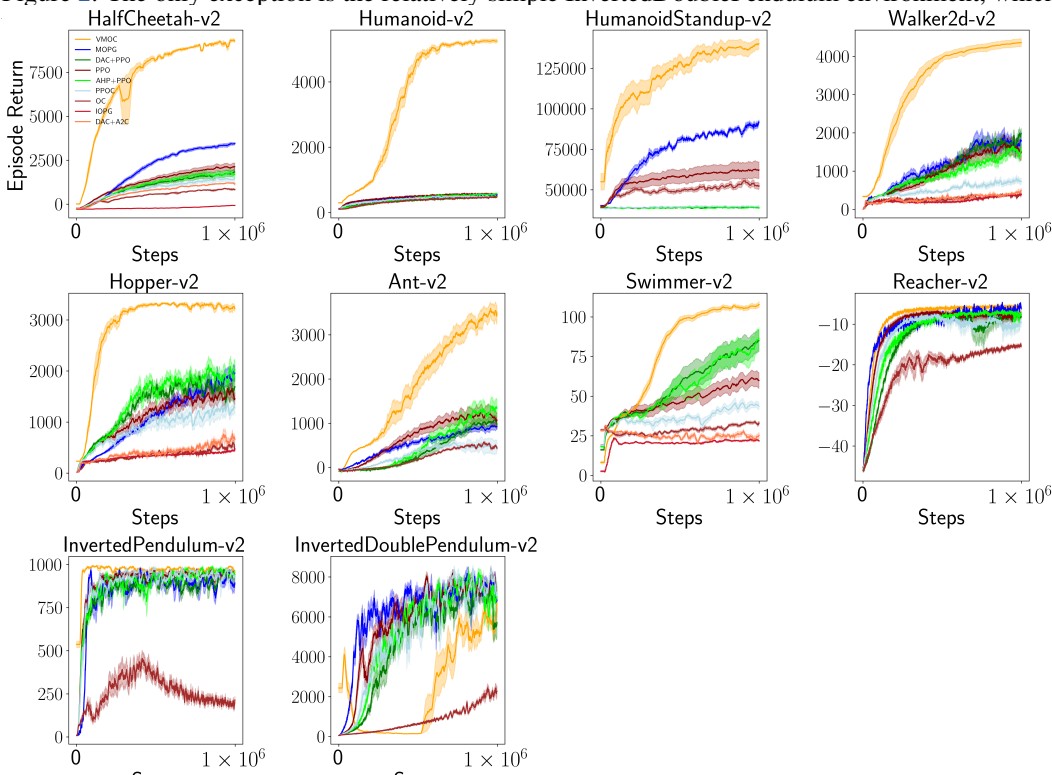

**Figure 2:** Experiments on Mujoco Environments. Curves are averaged over 10 independent runs with different random seeds and smoothed by a sliding window of size 20. Shaded regions indicate standard deviations.

Notably, VMOC exhibits superior performance on the Humanoid-v2 and HumanoidStandup-v2 environments. These environments are characterized by a large state space ($\mathcal{S} \in \mathbb{R}^{376}$) and action space ($\mathcal{A} \in \mathbb{R}^{17}$), whereas other environments typically have state dimensions less than 20 and action dimensions less than 5. The enhanced performance of VMOC in these environments can be attributed to its maximum entropy capability: in large state-action spaces, the agent must maximize rewards while exploring a diverse set of state-action pairs. Maximum likelihood methods tend to quickly saturate with early rewarding observations, leading to the selection of low-entropy options that converge to local optima.

A particularly relevant comparison is with the Markovian Option Policy Gradient (MOPG) [35], as both VMOC and MOPG are developed based on HiT-MDPs and employ option embeddings. Despite being derived under the maximum entropy framework, MOPG utilizes an on-policy gradient descent approach. Our experimental results show that VMOC's performance surpasses that of MOPG, highlighting the limitations of on-policy methods, which suffer from shortsighted rollout lengths and quickly saturate to early high-reward observations. In contrast, VMOC's variational off-policy approach effectively utilizes the maximum entropy framework by ensuring better exploration and stability across the learning process. Additionally, the off-policy nature of VMOC allows it to reuse samples from a replay buffer, enhancing sample efficiency and promoting greater diversity in the learned policies. This capability leads to more robust learning, as the algorithm can leverage a broader range of experiences to improve policy optimization.

## 6 Related Work

The VMOC incorporates three key ingredients: the option framework, a structural variational inference based off-policy algorithm and latent variable policies. We review prior works that draw

on some of these ideas in this section. The options framework [47] offers a promising approach for discovering and reusing temporal abstractions, with options representing temporally abstract skills. Conventional option frameworks [39], typically developed under the maximum likelihood (MLE) framework with few constraints on options behavior, often suffer from the option degradation problem [32, 4]. This problem occurs when options quickly saturate with early rewarding observations, causing a single option to dominate the entire policy, or when options switch every timestep, maximizing policy at the expense of skill reuse across tasks. On-policy option learning algorithms [4, 3, 52, 34, 35] aim to maximize expected return by adjusting policy parameters to increase the likelihood of high-reward option trajectories, which often leads to focusing on low-entropy options. Several techniques [20, 21, 23] have been proposed to enhance on-policy algorithms with entropy-like extrinsic rewards as regularizers, but these often result in biased optimal trajectories. In contrast, the maximum entropy term in VMOC arises naturally within the variational framework and provably converges to the optimal trajectory.

Although several off-policy option learning algorithms have been proposed [10, 43, 45, 50], these typically focus on improving sample efficiency by leveraging the control as inference framework. Recent works [45] aim to enhance sample efficiency by inferring and marginalizing over options, allowing all options to be learned simultaneously. Wulfmeier et al. [50] propose off-policy learning of all options across every experience in hindsight, further boosting sample efficiency. However, these approaches generally lack constraints on options behavior. A closely related work [33] also derives a variational approach under the option framework; however, it is based on probabilistic graphical model that we believe are incorrect, potentially leading to convergence issues. Additionally, our algorithm enables learning options as latent embeddings, a feature not present in their approach.

Recently, several studies have extended the maximum entropy reinforcement learning framework to discover skills by incorporating additional latent variables. One class of methods [22, 17] maintains latent variables constant over the duration of an episode, providing a time-correlated exploration signal. Other works [19, 51] focus on discovering multi-level action abstractions that are suitable for repurposing by promoting skill distinguishability, but they do not incorporate temporal abstractions. Studies such as [38, 1, 8] aim to discover temporally abstract skills essential for exploration, but they predefine their temporal resolution. In contrast, VMOC learns temporal abstractions as embeddings in an end-to-end data-driven approach with minimal prior knowledge encoded in the framework.

# 7 Conclusion

In this paper, we have introduced the Variational Markovian Option Critic (VMOC), a novel off-policy algorithm designed to address the challenges of ineffective exploration, sample inefficiency, and computational complexity inherent in the conventional option framework for hierarchical reinforcement learning. By integrating a variational inference framework, VMOC leverages maximum entropy as intrinsic rewards to promote the discovery of diverse and effective options. Additionally, by employing low-cost option embeddings instead of traditional, computationally expensive option triples, VMOC enhances both scalability and expressiveness. Extensive experiments in challenging Mujoco environments demonstrate that VMOC significantly outperforms existing on-policy and off-policy option variants, validating its effectiveness in learning coherent and diverse option sets suitable for complex tasks. This work advances the field of hierarchical reinforcement learning by providing a robust, scalable, and efficient method for learning temporally extended actions.

# 8 Limitations

Due to limited computing resources, we did not conduct an ablation study of VMOC. Additionally, the temperature parameter was fixed in our experiments, whereas an automatically tuned parameter could potentially enhance performance (see SAC [19]). While our baselines focus on option variants, a thorough comparison to other off-policy algorithms is also worth investigating. It is particularly important to explore whether VMOC exhibits performance improvements in scalability when the number of option embeddings is significantly increased. These investigations are left for future work.

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

# A  Proofs

## A.1  Theorem 1

**Theorem 1** (Convergence Theorem for Structured Variational Policy Iteration). *Let $\tau$ be the latent variable and $\mathcal{E}$ be the observed variable. Define the variational distribution $q(\tau)$ and the log-likelihood $\log P(\mathcal{E})$. Let $M : q^{[k]} \to q^{[k+1]}$ represent the mapping defined by the EM steps inference update, so that $q^{[k+1]} = M(q^{[k]})$. The likelihood function increases at each iteration of the variational inference algorithm until the conditions for equality are satisfied and a fixed point of the iteration is reached:*

$$\log P(\mathcal{E} \mid q^{[k+1]}) \geq \log P(\mathcal{E} \mid q^{[k]}), \text{ with equality if and only if}$$

$$\mathcal{L}(q^{[k+1]}, P) = \mathcal{L}(q^{[k]}, P)$$

*and*

$$D_{KL}(q^{[k+1]}(\tau) \parallel P(\tau \mid \mathcal{E})) = D_{KL}(q^{[k]}(\tau) \parallel P(\tau \mid \mathcal{E})).$$

477 *Proof.* Let $\tau$ be the latent variable and $\mathcal{E}$ be the observed variable. Define the evidence lower bound
478 (ELBO) as $\mathcal{L}(q, P)$ and the Kullback-Leibler divergence as $\mathrm{D}_{\mathrm{KL}}(q \parallel P)$, where $q(\tau)$ approximates
479 the posterior distribution and $P(\mathcal{E} \mid \tau)$ is the likelihood.

480 The log-likelihood function $\log P(\mathcal{E})$ can be decomposed as:

$$\log P(\mathcal{E}) = \mathcal{L}(q, P) + \mathrm{D}_{\mathrm{KL}}(q(\tau) \parallel P(\tau \mid \mathcal{E})),$$

481 where

$$\mathcal{L}(q, P) = \mathbb{E}_{q(\tau)} \left[ \log P(\mathcal{E}, \tau) - \log q(\tau) \right]$$

482 and

$$\mathrm{D}_{\mathrm{KL}}(q(\tau) \parallel P(\tau \mid \mathcal{E})) = \mathbb{E}_{q(\tau)} \left[ \log \frac{q(\tau)}{P(\tau \mid \mathcal{E})} \right].$$

483 Let $M : q^{[k]} \to q^{[k+1]}$ represent the mapping defined by the variational inference update, so that
484 $q^{[k+1]} = M(q^{[k]})$. If $q^*$ is a variational distribution that maximizes the ELBO, so that $\log P(\mathcal{E} \mid$
485 $q^*) \geq \log P(\mathcal{E} \mid q)$ for all $q$, then $\log P(\mathcal{E} \mid M(q^*)) = \log P(\mathcal{E} \mid q^*)$. In other words, the
486 maximizing distributions are fixed points of the variational inference algorithm. Since the likelihood
487 function is bounded (for distributions of practical interest), the sequence of variational distributions
488 $q^{[0]}, q^{[1]}, \ldots, q^{[k]}$ yields a bounded nondecreasing sequence $\log P(\mathcal{E} \mid q^{[0]}) \leq \log P(\mathcal{E} \mid q^{[1]}) \leq$
489 $\cdots \leq \log P(\mathcal{E} \mid q^{[k]}) \leq \log P(\mathcal{E} \mid q^{[k]})$ which must converge as $k \to \infty$.

490 $\square$

## A.2 Theorem 2

492 **Theorem 2** (Soft Option Policy Iteration Theorem). *Repeated optimizing $\mathcal{L}$ and $\mathrm{D}_{\mathrm{KL}}$ defined in*
493 *Eq. 10 from any $\pi_0^A, \pi_0^O \in \Pi$ converges to optimal policies $\pi^{A*}, \pi^{O*}$ such that $Q_O^{soft*}[\mathbf{s}_t, \mathbf{o}_t] \geq$*
494 *$Q_O^{soft}[\mathbf{s}_t, \mathbf{o}_t]$ and $Q_A^{soft*}[\mathbf{s}_t, \mathbf{o}_t, \mathbf{a}_t] \geq Q_A^{soft}[\mathbf{s}_t, \mathbf{o}_t, \mathbf{a}_t]$, for all $\pi_0^A, \pi_0^O \in \Pi$ and $(\mathbf{s}_t, \mathbf{a}_t, \mathbf{o}_t) \in$*
495 *$\mathcal{S} \times \mathcal{A} \times \mathcal{O}$, assuming $|\mathcal{S}| < \infty$, $|\mathcal{O}| < \infty$, $|\mathcal{A}| < \infty$.*

496 *Proof.* Define the entropy augmented reward as $r^{soft}(\mathbf{s}_t, \mathbf{a}_t) = r(\mathbf{s}_t, \mathbf{a}_t) + \mathcal{H}[\pi^A]$ and
497 $f^{soft}(\mathbf{o}_t, \mathbf{s}_t, \mathbf{a}_t, \mathbf{o}_{t-1}) = f(\mathbf{o}_t, \mathbf{s}_t, \mathbf{a}_t, \mathbf{o}_{t-1}) + \mathcal{H}[\pi^O]$ and rewrite Bellman Backup functions as,

$$Q_O[\mathbf{s}_t, \mathbf{o}_t] = f^{soft}(\cdot) + \mathbb{E}_{\mathbf{a}_t \sim \pi^A} \left[ Q_A[\mathbf{s}_t, \mathbf{o}_t, \mathbf{a}_t] \right],$$
$$Q_A[\mathbf{s}_t, \mathbf{o}_t, \mathbf{a}_t] = r^{soft}(s, a) + \mathbb{E}_{\mathbf{s}_{t+1} \sim P(\mathbf{s}_{t+1} | \mathbf{s}_t, \mathbf{a}_t)} \left[ \mathbb{E}_{\mathbf{o}_{t+1} \sim \pi^O} \left[ Q_O[\mathbf{s}_{t+1}, \mathbf{o}_{t+1}] \right] \right]$$

498 We start with proving the convergence of soft option policy evaluation. As with the standard Q-
499 function and value function, we can relate the Q-function at a future state via a *Bellman Operator*
500 $\mathcal{T}^{soft}$. The option-action value function satisfies the Bellman Operator $\mathcal{T}^{soft}$

$$\mathcal{T}^{soft} Q_A[\mathbf{s}_t, \mathbf{o}_t, \mathbf{a}_t] = \mathbb{E}[G_t | \mathbf{s}_t, \mathbf{o}_t, \mathbf{a}_t]$$
$$= r^{soft}(s, a) + \gamma \sum_{\mathbf{s}_{t+1}} P(\mathbf{s}_{t+1} | \mathbf{s}_t, \mathbf{a}_t) Q_O[\mathbf{s}_{t+1}, \mathbf{o}_t],$$

501 As with the standard convergence results for policy evaluation [46], by the definition of $\mathcal{T}^{soft}$ (Eq. 11)
502 the option-action value function $Q_A^{\pi_A}$ is a fixed point.

503 To prove the $\mathcal{T}^{soft}$ is a contraction, define a norm on $V$-values functions $V$ and $U$

$$\|V - U\|_\infty \triangleq \max_{\bar{s} \in \bar{S}} |V(\bar{s}) - U(\bar{s})|. \tag{11}$$

504 where $\bar{s} = \{s, o\}$.

505 By recurssively apply the Hidden Temporal Bellman Operator $\mathcal{T}^{soft}$, we have:

$$Q_O[\mathbf{s}_t, \mathbf{o}_{t-1}] = \mathbb{E}[G_t|\mathbf{s}_t, \mathbf{o}_{t-1}] = \sum_{\mathbf{o}_t} P(\mathbf{o}_t|\mathbf{s}_t, \mathbf{o}_{t-1}) Q_O[\mathbf{s}_t, \mathbf{o}_t]$$

$$= \sum_{\mathbf{o}_t} P(\mathbf{o}_t|\mathbf{s}_t, \mathbf{o}_{t-1}) \sum_{\mathbf{a}_t} P(\mathbf{a}_t|\mathbf{s}_t, \mathbf{o}_t) \left[ r(s,a) + \gamma \sum_{\mathbf{s}_{t+1}} P(\mathbf{s}_{t+1}|\mathbf{s}_t, \mathbf{a}_t) Q_O[\mathbf{s}_{t+1}, \mathbf{o}_t] \right]$$

$$= r(s,a) + \gamma \sum_{\mathbf{o}_t} P(\mathbf{o}_t|\mathbf{s}_t, \mathbf{o}_{t-1}) \sum_{\mathbf{a}_t} P(\mathbf{a}_t|\mathbf{s}_t, \mathbf{o}_t) \sum_{\mathbf{s}_{t+1}} P(\mathbf{s}_{t+1}|\mathbf{s}_t, \mathbf{a}_t) Q_O[\mathbf{s}_{t+1}, \mathbf{o}_t]$$

$$= r(s,a) + \gamma \sum_{\mathbf{o}_t, \mathbf{s}_{t+1}} P(\mathbf{s}_{t+1}, \mathbf{o}_t|\mathbf{s}_t, \mathbf{o}_{t-1}) Q_O[\mathbf{s}_{t+1}, \mathbf{o}_t]$$

$$= r(s,a) + \gamma E_{\mathbf{s}_{t+1}, \mathbf{o}_t} \left[ Q_O[\mathbf{s}_{t+1}, \mathbf{o}_t] \right] \tag{12}$$

Therefore, by applying Eq. 12 to $V$ and $U$ we have:

$$\|T^\pi V - T^\pi U\|_\infty$$
$$= \max_{\bar{s} \in \bar{S}} \left| \gamma E_{\mathbf{s}_{t+1}, \mathbf{o}_t} \left[ Q_O[\mathbf{s}_{t+1}, \mathbf{o}_t] \right] - \gamma E_{\mathbf{s}_{t+1}, \mathbf{o}_t} \left[ U[\mathbf{s}_{t+1}, \mathbf{o}_t] \right] \right|$$
$$= \gamma \max_{\bar{s} \in \bar{S}} E_{\mathbf{s}_{t+1}, \mathbf{o}_t} \left[ \left| Q_O[\mathbf{s}_{t+1}, \mathbf{o}_t] - U[\mathbf{s}_{t+1}, \mathbf{o}_t] \right| \right]$$
$$\leq \gamma \max_{\bar{s} \in \bar{S}} E_{\mathbf{s}_{t+1}, \mathbf{o}_t} \left[ \gamma \max_{\bar{s} \in \bar{S}} \left| Q_O[\mathbf{s}_{t+1}, \mathbf{o}_t] - U[\mathbf{s}_{t+1}, \mathbf{o}_t] \right| \right]$$
$$\leq \gamma \max_{\bar{s} \in \bar{S}} |V[\bar{s}] - U[\bar{s}]|$$
$$= \gamma \|V - U\|_\infty \tag{13}$$

Therefore, $\mathcal{T}^{soft}$ is a contraction. By the fixed point theorem, assuming that throughout our computation the $Q_A[\cdot, \cdot]$ and $Q_O[\cdot]$ are bounded and $\mathbb{A} < \infty$, the sequence $Q_A^k$ defined by $Q_A^{k+1} = \mathcal{T}^{soft} Q_A^k$ will converge to the option-action value function $Q_A^{\pi_A}$ as $k \to \infty$.

The convergence results of and the Soft Option Policy Improvement Theorem then follows conventional Soft Policy Improvement Theorem Theorem 1. Consequently, the Soft Option Policy Iteration Theorem follows directly from these results.

$\square$

## A.3 Derivation of Eq. 10

$$\mathcal{L}(q(\tau), P(\tau, \mathcal{E}_{1:T}^A, \mathcal{E}_{1:T}^O)) = \mathbb{E}_{q(\tau)}[\log P(\tau, \mathcal{E}_{1:T}^A, \mathcal{E}_{1:T}^O) - \log q(\tau)]$$
$$= \mathbb{E}_{q(\tau)}[\log P(\tau|\mathcal{E}_{1:T}^A, \mathcal{E}_{1:T}^O) + \log P(\mathcal{E}_{1:T}^A, \mathcal{E}_{1:T}^O) - \log q(\tau)]$$
$$= \mathbb{E}_{q(\tau)}[\log P(\tau|\mathcal{E}_{1:T}^A, \mathcal{E}_{1:T}^O) - \log q(\tau)] + \mathbb{E}_{q(\tau)} \log P(\mathcal{E}_{1:T}^A, \mathcal{E}_{1:T}^O)$$
$$= \mathbb{E}_{q(\tau)}\left[\frac{\log P(\tau|\mathcal{E}_{1:T}^A, \mathcal{E}_{1:T}^O)}{\log q(\tau)}\right] + \log P(\mathcal{E}_{1:T}^A, \mathcal{E}_{1:T}^O)$$
$$= -D_{\mathrm{KL}}(\log q(\tau) \,\|\, \log P(\tau|\mathcal{E}_{1:T}^A, \mathcal{E}_{1:T}^O)) + \log P(\mathcal{E}_{1:T}^A, \mathcal{E}_{1:T}^O)$$

## A.4 Theorem 3

**Theorem 3** (Convergence Theorem for Variational Markovian Option Policy Iteration). *Let $\tau$ be the latent variable and $\mathcal{E}^A, \mathcal{E}^O$ be the ground-truth optimality variables. Define the variational distribution $q(\tau)$ and the true log-likelihood of optimality $\log P(\mathcal{E}^A, \mathcal{E}^O)$. iterates according to the update rule $q^{k+1} = \arg\max_q \mathcal{L}(q(\tau), P(\tau, \mathcal{E}_{1:T}^A, \mathcal{E}_{1:T}^O))$ converges to the maximum value bounded by the data log-likelihood.*

*Proof.* The objective is to maximize the ELBO with respect to the policy $q$. Formally, this can be written as:

$$q^{k+1} = \arg\max_q \mathcal{L}(q, P).$$

Suppose we $q$ is a neural network function approximator, assuming the continuity and differentiability of $q$ with respect to its parameters. Using stochastic gradient descent (SGD) to optimize the parameters guarantees that the ELBO increases, such that $\mathcal{L}(q^{k+1}, P) \geq \mathcal{L}(q^k, P)$.

Rearranging Eq. 10 we get:

$$
\begin{aligned}
D_{\mathrm{KL}}(q^{k+1}(\tau)||P(\tau|\mathcal{E}_{1:T}^A, \mathcal{E}_{1:T}^O)) &= -L(q^{k+1}(\tau), P(\tau, \mathcal{E}_{1:T}^A, \mathcal{E}_{1:T}^O)) + \log P(\mathcal{E}_{1:T}^A, \mathcal{E}_{1:T}^O) \\
&\leq -L(q^k(\tau), P(\tau, \mathcal{E}_{1:T}^A, \mathcal{E}_{1:T}^O)) + \log P(\mathcal{E}_{1:T}^A, \mathcal{E}_{1:T}^O) \\
&= D_{\mathrm{KL}}(q^k(\tau)||P(\tau|\mathcal{E}_{1:T}^A, \mathcal{E}_{1:T}^O))
\end{aligned}
$$

Thus, each SGD update not only potentially increases the ELBO but also decreases the KL divergence, moving $q$ closer to $P$. Given the properties of SGD and assuming appropriate learning rates and sufficiently expressive neural network architectures, the sequence $\{q^k\}$ converges to a policy $q^*$ that minimizes the KL divergence to the true posterior. $\square$

# B   VMOC Algorithm

---
**Algorithm 1** VMOC Algorithm

---
1: Initialize parameter vectors $\psi^A$, $\psi^O$, $\theta^O$, $\theta^A$
2: **for** each epoch **do**
3:     Collect trajectories $\{\mathbf{o}_{t-1}, \mathbf{s}_t, \mathbf{a}_t, \mathbf{o}_t\}$ into the replay buffer
4:     **for** each gradient step **do**
5:         Update $Q_{\psi_i^A}^{soft}$: $\psi_i^A \leftarrow \psi_i^A - \eta_{Q^A} \nabla J_{Q_{\psi_i^A}^{soft}}$ for $i \in \{1, 2\}$
6:         Update $Q_{\psi_i^O}^{soft}$: $\psi_i^O \leftarrow \psi_i^O - \eta_{Q^O} \nabla J_{Q_{\psi_i^O}^{soft}}$ for $i \in \{1, 2\}$
7:         Update $\pi_{\theta^O}^O$: $\theta^O \leftarrow \theta^O - \eta_{\pi^O} \nabla J_{\pi^O}$
8:         Update $\pi_{\theta^A}^A$: $\theta^A \leftarrow \theta^A - \eta_{\pi^A} \nabla J_{\pi^A}$
9:         Update target networks: $\bar{\psi}^A \leftarrow \sigma\psi^A + (1-\sigma)\bar{\psi}^A$, $\bar{\psi}^O \leftarrow \sigma\psi^O + (1-\sigma)\bar{\psi}^O$
10:         Update temperature factors: $\alpha^O \leftarrow \alpha^O - \eta_{\alpha^O}\nabla J_{\alpha^O}$, $\alpha^A \leftarrow \alpha^A - \eta_{\alpha^A}\nabla J_{\alpha^A}$
11:     **end for**
12: **end for**

---

# C   Implementation Details

## C.1   Hyperparameters

In this section we summarize our implementation details. For a fair comparison, all baselines: MOPG [35], DAC+PPO [52], AHP+PPO [32], PPOC [27], OC [4] and PPO [41] are from DAC's open source Github repo: https://github.com/ShangtongZhang/DeepRL/tree/DAC. Hyperparameters used in DAC [52] for all these baselines are kept unchanged.

**VMOC Network Architecture:** We use Pytorch to build neural networks. Specifically, for option embeddings, we use an embedding matrix $\boldsymbol{W}_S \in \mathbb{R}^{4 \times 40}$ which has 4 options (4 rows) and an embedding size of 40 (40 columns). For layer normalization we use Pytorch's built-in function LayerNorm [2]. For Feed Forward Networks (FNN), we use a 2 layer FNN with ReLu function as activation function with input size of state-size, hidden size of $[256, 256]$, and output size of action-dim neurons. For Linear layer, we use built-in Linear function[3] to map FFN's outputs to 4 dimension.

---
[2]https://pytorch.org/docs/stable/generated/torch.nn.LayerNorm.html
[3]https://pytorch.org/docs/stable/generated/torch.nn.Linear.html

Each dimension acts like a logit for each skill and is used as density in Categorical distribution[4]. For both action policy and critic module, FFNs are of the same size as the one used in the skill policy.

**Preprocessing:** States are normalized by a running estimation of mean and std.

**Hyperparameters for all on-policy option variants:** For a fair comparison, we use exactly the same parameters of PPO as DAC . Specifically:

- Optimizer: Adam with $\epsilon = 10^{-5}$ and an initial learning rate $3 \times 10^{-4}$
- Discount ratio $\gamma$: 0.99
- GAE coefficient: 0.95
- Gradient clip by norm: 0.5
- Rollout length: 2048 environment steps
- Optimization epochs: 10
- Optimization batch size: 64
- Action probability ratio clip: 0.2

**Computing Infrastructure:** We conducted our experiments on an Intel® Core™ i9-9900X CPU @ 3.50GHz with a single thread and process with PyTorch.

---

[4]https://github.com/pytorch/pytorch/blob/master/torch/distributions/categorical.py

