# OpenReview forum: "Learning Variational Temporal Abstraction Embeddings in Option-Induced MDPs"
_NeurIPS.cc/2024/Conference — Submitted to NeurIPS 2024_

### Official Review · Reviewer_AsEd · 2024-06-20

**Soundness:** 3
**Presentation:** 2
**Contribution:** 1
**Rating:** 3
**Confidence:** 4

**Summary:**

The paper presents an off-policy hierarchical RL method, based on the HiT-MDP formulation of a Semi-MDP. The HiT-MDP formulation treats the option $o$ as an extension of the original state $s$ (which can be chosen by an extended action), and combines initialization-, termination- and option-policy in a single Markovian master policy $p(o\_{t}|s\_t,o\_{t-1})$. The policy in the extended state-action space, thus, decomposes into the high-level and low-level policies, $p(o\_{t}, a\_t | s\_t, o\_{t-1}) = p(o\_{t} | s\_t, o\_{t-1}) p(a\_t | s\_t, o\_{t})$, which can be trained using standard RL algorihms.
Compared to the prior work, the paper makes the following contributions:
- Whereas previously PPO was used for reinforcement learning, the paper proposes to use SAC, resulting in improved sample efficiency
- The paper motivates the algorithm from a control-as-inference perspective

**Strengths:**

The proposed method seems to be technically sound, and using off-policy agents for HiT-MDPs seems sensible. (Quality)

The provided code clarifies the implementation which helps reproducibility. (Quality)

The presentation is mostly clear. (Clarity)

Applying an off-policy agents to HiT-MDPs seems to be novel and effective (Origingality, Significance)

**Weaknesses:**

Originality
-----------
One of the main weaknesses of the submission is the limited novelty. Replacing the PPO agent of MOPG by a SAC agent seems to be straightforward, so this contribution is quite incremental. Indeed, the authors of HiT-MDP stated, that their ELBO "can easily be extended to a SAC-like algorithm" [35]. Furthermore, given that MaxEnt-RL was already derived from a control-as-inference perspective, deriving the special case of an HiT-MDP using this technique does not seem to be significant contribution either. I also don't see the value of this derivation that would justify devoting so much space on it; couldn't we just argue that we apply SAC to such particular form of an MDP?


Quality
---------
The experimental evaluation seems to be another weakness of the submission. While the method is evaluated on a reasonable number of MuJoCo environments, where it outperforms a reasonable number of baselines, the choice of baselines is not convincing because it looks like the method is only compared to on-policy algorithm. The submission claims that there method "significantly outperforms existing on-policy and off-policy option variants", but it is not clear to me to which off-policy baselines this claim refers to. It would be important to focus to flat and hierarchical off-policy methods in the experiments, such as [19], [50], [33] and Hao et. al (2023).  Furthermore, the choice of environments is not convincing, because it does not include more challenging long-horizon tasks that are typically used for evaluating HRL methods, such as Ant-Maze. While the performance on the standard locomotion environments is reasonable, the reported numbers don't seem to improve on the SOTA of flat-RL methods.

The paper does not discuss the hyperparameter search although it states in the questionnary  that these details are provided in the main content and the appendix.

The paper argues that it did not perform any ablations due to limited computational resources. However, I don't find this argument very convincing, since the experiments are performed on simple vision-free locomotion tasks, that can be run on standard workstation, not even requiring any GPU. Ablations on the number of options would be very useful.



Clarity
---------
I found the background material on control-as-inference a bit confusing. In particular, line 106 which states states policy improvement constitues an M-Step of an EM algorithm that *maximizes* the KL towards $P(\tau|\mathcal{E})$. I don't think any practical algorithm involves such maximization, since the optimum would correspond to a delta distribution on the least-likely trajectory. (

Visually, the presentation is rather bad. Figures are not on the top, and in particular Fig. 2 seems to hide some text, since the sentence in line 271 ends with ", which". Fig. 2 itself could be improved by increasing the plot sizes (there are some unnecessary white spaces) and by making the legend more readable.


Significance
-----------------
While I think that the proposed combination of the HiT-MDP formulation and SAC is somewhat interesting, the submission does not provide a convincing argument for the method. When should I use it, instead of existing (hierarchical or flat) methods?

References
----------
* Hao, C., Weaver, C., Tang, C., Kawamoto, K., Tomizuka, M., & Zhan, W. (2023). Skill-critic: Refining learned skills for reinforcement learning. arXiv preprint arXiv:2306.08388.

**Questions:**

* Line 227 mentions that the targets for the option Q-function are computed using actions from the replay buffer because estimating the expectation with respect to samples from the policy would be intractable. I don't understand this: Why can't we just sample from the policy?

* Which of the baselines where off-policy?

**Limitations:**

The limitations are adequately discussed and I don't have any concerns regarding negative societal impact of the work.

---

### Official Review · Reviewer_8cLu · 2024-07-11

**Soundness:** 2
**Presentation:** 2
**Contribution:** 1
**Rating:** 4
**Confidence:** 3

**Summary:**

This paper proposes the Variational Markovian Option Critic (VMOC), an off-policy algorithm for hierarchical reinforcement learning. VMOC aims to address exploration inefficiency and update instability in existing methods. Key contributions include: 1. Use of variational inference for update stabilization 2. Low-cost option embeddings for improved scalability. The authors evaluate VMOC on Mujoco environments, comparing it to other on-policy and off-policy methods. They report improved performance in learning option sets for complex tasks.

**Strengths:**

1. The paper is well-written, and the proposed method is theoretically justified.
2. The empirical evaluations show favorable results compared with existing methods.

**Weaknesses:**

1. Very similar ideas of the variational option framework have been proposed in [33] (off-policy) and [35] (on-policy). While [35] proposes an on-policy version, its off-policy version is also straightforward to deduce following [ref1]. The use of option embeddings is following [35].
2. The empirical evaluations are very limited; there is no ablative evaluation reported, which makes it hard to determine the contribution of the proposed method to the overall performance gain over various baselines.

References:
[ref1] Levine, Sergey. "Reinforcement learning and control as probabilistic inference: Tutorial and review." arXiv preprint arXiv:1805.00909 (2018).

**Questions:**

The authors claim that [33] is based on an "incorrect" probabilistic graphical model. I wonder if the authors could elaborate on this claim.

**Limitations:**

The empirical evaluations, especially ablation studies, are somewhat limited in scope.

---

### Official Review · Reviewer_w5JT · 2024-07-15

**Soundness:** 3
**Presentation:** 3
**Contribution:** 3
**Rating:** 6
**Confidence:** 3

**Summary:**

The paper introduces the Variational Markovian Option Critic (VMOC) which combines variation policy iteration and the option critic. VMOC also modifies HiT-MDPs, where options are represented as latent embeddings rather than triples of (init states, policy, termination condition), to the off-policy setting. The paper performs comparisons to option-based methods and PPO on 10 Mujoco environments.

**Strengths:**

1. The paper is well-written and easy-to-read. The figures clearly highlight the performance of the method. The translation from theory to the practical algorithm is well detailed.

2. The advantage in sample-efficiency over other option methods and PPO is clearly seen in Figure 2 across Mujoco environments. In fact, this gain looks to be in atleast two orders-of-magnitude (of fewer steps required by VMOC) which is amazing. The underlying MaxEnt objective in VMOC appears to be very useful with exploration in the high-dim mujoco envs.

**Weaknesses:**

1. It is not clear if this gain in sample-efficiency will transfer to discrete environments or is somehow applicable only in continuous envs. Perhaps the authors can perform comparisons on Atari or Procgen to demonstrate the same? It would be great if the authors could also discuss the changes in the algorithm in the discrete and continuous settings (perhaps such as the sampling of a_t from the replay buffer?)

2. It is unclear if all methods use the same number of options (e.g. the value used in VMOC appears to be 4). A clear ablation of various design choices like number of options would help demonstrate that VMOC is thoroughly better than the other option methods and is not brittle to hyperparameter choice.

The analysis of the actual options learnt is also missing (this is for example seen in the option critic paper). This, alongside an analysis of the number of options, is crucial to understand if the method is actual learning composed actions that are further composable and generalizable or degenerating to something simple like learning the action primitives (although the latter would apply more to a discrete rather than continuous env).

3. Minor comment: The location of Theorem 1 in the preliminaries makes it unclear if it is a contribution of the authors or well-known statement. Perhaps the authors can clarify?

4. Another minor comment: It would be great if the authors could discuss other ways of combining options in the related work such as in [1] and [2].

[1] The Option Keyboard: Combining Skills in Reinforcement Learning, Barreto et al, NeurIPS 2019

[2] Exploring with Sticky Mittens: Reinforcement Learning with Expert Interventions via Option Templates, Dutta et al, CoRL 2022

**Questions:**

Please see Weaknesses.

**Limitations:**

The authors have addressed limitations.

---

### Official Review · Reviewer_WKv9 · 2024-07-18

**Soundness:** 3
**Presentation:** 1
**Contribution:** 2
**Rating:** 4
**Confidence:** 3

**Summary:**

This paper introduces e Variational Markovian Option Critic (VMOC), which learns actions and options simultatenously. They build upon the Hidden Temporal Markovian Decision Process (HiT-MDP) [1] to build a novel off-policy algorithm that utilizes entropy augmented rewards. Their method learns options’ embedding vectors (rather than conventional option tuples utilized in Semi-MDP [2]). They benchmark the learning performance of their method against several competitors on many classic control benchmark environments.

*References*
1. Li, C., Song, D., & Tao, D. (2023). Hit-MDP: learning the SMDP option framework on MDPs with hidden temporal embeddings. In The Eleventh International Conference on Learning Representations.
2. Sutton, R. S., Precup, D., & Singh, S. (1999). Between MDPs and semi-MDPs: A framework for temporal abstraction in reinforcement learning. Artificial intelligence, 112(1-2), 181-211.

**Strengths:**

- Extensive comparison against 8 competitor algorithms on 10 benchmark tasks
- A novel Soft Option Policy Iteration Theorem

**Weaknesses:**

The paper does offer a potentially interesting contribution to the wider research community. But it is held back by the lack of clarity and polish in writing. For example, two glaring signs of a hasty submission:
1. Sec 4 and Sec 5 are both titled experiments. Sec 4 is only 1 paragraph, and it essentially repeats the same information from the introductory paragraphs of Sec 5
2. In Sec. 5, line 271 just trails off without completion. I believe the authors moved around the images to correct for vertical space and accidentally hid the text.

While the experimental results focus on learning curves, where VMOC does well, they fail to provide other relevant evaluation metrics:
1. What do the learned options look like? A good evaluation could follow Fig. 5 and Fig. 6 from [1]
2. How many options are learned? Digging through the appendix, it says that they learned 4 option vectors. This leads to another question: how do they choose the number of options to learn?
3. The VMOC algorithm listed in the appendix only describes the gradient update process. No details about action sampling or other hyper-parameter tuning are described here
4. The environments used are challenging for model-free RL algorithms. That said, they may not be satisfactory for showcasing the potential of learned options.


*References*
1. Li, C., Song, D., & Tao, D. (2023). Hit-MDP: learning the SMDP option framework on MDPs with hidden temporal embeddings. In The Eleventh International Conference on Learning Representations.

**Questions:**

- How are actions sampled? Do you use the reparametrization trick  [1] for sampling action? Is the update similar to SAC?
- How does your algorithm compare against SAC? Looking at the gradient update rules, one could argue that it is fairer to compare VMOC against SAC than PPO. My suspicion is that SAC would perform just as well as VMOC
- Why mujoco environments for showcasing options? The true benefit of options would be seen in environments that could benefit from hierarchical policies or composition of policies. Do the authors consider environments from [Meta-World](https://meta-world.github.io/) or the dog fetch or ant  fetch environments in [DM Control Suite](https://github.com/google-deepmind/dm_control/tree/main/dm_control/suite)

*References*
1. Kingma, D. P., & Welling, M. (2013). Auto-encoding variational bayes. ArXiv Preprint ArXiv:1312.6114.
2. Haarnoja, T., Zhou, A., Abbeel, P., & Levine, S. (2018, July). Soft actor-critic: Off-policy maximum entropy deep reinforcement learning with a stochastic actor. In International conference on machine learning (pp. 1861-1870). PMLR.

**Limitations:**

Much like Soft Actor-Critic, this work develops a novel Soft Option Critic style algorithm. I believe this line of work is very interesting and potentially impactful in the near future. However, their current draft is not well-written and hard to follow. Their experimental evaluation is also insufficient.

---

### Author Rebuttal · Authors · 2024-08-07

We deeply appreciate reviewers' efforts. We will incoporate reviewers suggestions in our next version.

---

### Decision · Program_Chairs · 2024-09-25

**Decision:**

Reject

**Comment:**

The reviewers agree that there's much room for improvement, from the stated contributions to the empirical protocol to better positioning the paper in the literature. The fact that no thorough response was provided to the reviewers, but only an acknowledgement that the paper will be improved in its next version, made the decision quite obvious.